# A High Respiratory Drive Is Associated with Weaning Failure in Patients with COVID-19-Associated Acute Respiratory Distress Syndrome: The Role of the Electrical Activity of the Diaphragm

**DOI:** 10.3390/jcm13041120

**Published:** 2024-02-16

**Authors:** Stefano Muttini, Jacopo Jona Falco, Ilmari Cuevas Cairo, Michele Umbrello

**Affiliations:** 1Neuroscience Intensive Care Unit, San Carlo Borromeo Hospital, ASST Santi Paolo e Carlo, 20151 Milano, Italy; stefano.muttini@asst-santipaolocarlo.it (S.M.); dr@jonafalco.it (J.J.F.); 2Department of Anaesthesia and Intensive Care Unit, San Carlo Borromeo Hospital, ASST Santi Paolo e Carlo, 20151 Milano, Italy; ilmari.cuevascairo1@asst-santipaolocarlo.it; 3Department of Intensive care and Anaesthesia, Ospedale Civile di Legnano, ASST Ovest Milanese, 20025 Legnano, Italy

**Keywords:** critically ill patients, COVID-19, ARDS, respiratory drive, electrical activity of the diaphragm, weaning

## Abstract

Background: Mechanical ventilation is the main supportive treatment of severe cases of COVID-19-associated ARDS (C-ARDS). Weaning failure is common and associated with worse outcomes. We investigated the role of respiratory drive, assessed by monitoring the electrical activity of the diaphragm (EAdi), as a predictor of weaning failure. Methods: Consecutive, mechanically ventilated patients admitted to the ICU for C-ARDS with difficult weaning were enrolled. Blood gas, ventilator, and respiratory mechanic parameters, as well as EAdi, were recorded at the time of placement of EAdi catheter, and then after 1, 2, 3, 7, and 10 days, and compared between patients with weaning success and weaning failure. Results: Twenty patients were enrolled: age 66 (60–69); 85% males; PaO_2_/FiO_2_ at admission 148 (126–177) mmHg. Thirteen subjects (65%) were classified as having a successful weaning. A younger age (OR(95%CI): 0.02 (0.01–0.11) per year), a higher PaO_2_/FiO_2_ ratio (OR(95%CI): 1.10 (1.01–1.21) per mmHg), and a lower EAdi (OR(95%CI): 0.16 (0.08–0.34) per μV) were associated with weaning success. Conclusion: In critically ill patients with moderate–severe C-ARDS and difficult weaning from mechanical ventilation, a successful weaning was associated with a lower age, a higher oxygenation, and a lower respiratory drive, as assessed at the bedside via EAdi monitoring.

## 1. Introduction

While most people with COVID-19 experience mild symptoms, around 15% may develop a severe condition known as COVID-19-associated acute respiratory distress syndrome (C-ARDS). This is the primary reason for ICU admission and is linked to a poor outcome [1]. As with “classic” ARDS, the cornerstone of the treatment of C-ARDS lies in the early identification and management of the underlying cause, such as infection recognition and appropriate antimicrobial treatment [2]. This etiological therapy often needs to be associated with supportive treatments, which mainly consist of invasive and non-invasive mechanical ventilation [3,4] to ensure adequate gas exchange until acute lung disease is overcome [5]. However, mechanical ventilation is also associated with potential complications, such as ventilator-induced lung injury, ventilator-induced diaphragm dysfunction, and ventilation-associated pneumonia [6,7,8]. As prolonging the duration of ventilatory care places patients at risk of developing ventilation-related complications, the correct identification of the time window within which to begin the weaning phase from mechanical ventilation seems vital, as is the recognition of patients who will not yet benefit from weaning.

In the ICU setting, weaning success is commonly defined as the ability to maintain spontaneous breathing in the absence of ventilatory support for at least 48 h. Conversely, weaning failure consists of the failure of spontaneous breathing trial (SBT), resumption of ventilatory support following successful extubation, or death within 48 h following extubation [9,10]. Patients who fail initial weaning and require at least three SBTs or at least 7 days from the first SBT to successful weaning are recognized as patients with difficult weaning [10]. Previous studies in unselected ICU patients showed that up to 26% of patients have difficult weaning, and this was associated with increased mortality [11]; recent data on C-ARDS confirmed these findings [12].

Since the inspiratory muscles are supposed to uptake the bulk of the respiratory effort after discontinuation of mechanical ventilation, monitoring of the respiratory drive, i.e., the intensity of the output of the respiratory centers, which determines the mechanical output of the respiratory muscles, might be a useful tool to assess readiness for weaning [13]. The electrical activity of the diaphragm (EAdi) is the spatial and temporal sum of the electrical activities of all the motor units of the diaphragm [14]. EAdi is measured via transesophageal diaphragmatic electromyography by the EAdi tube, a peculiar nasogastric tube with the traditional lumen for enteral feeding and eight pairs of bipolar electrodes placed at the distal end [15,16]. EAdi is considered the best proxy for the activity of neural respiratory centers (i.e., the respiratory drive) currently available in clinical practice [17]. As long as the phrenic nerve and neuroventilatory coupling are intact, and the diaphragm is used as the main inspiratory muscle, the EAdi inspiratory peak was shown to correspond with respiratory drive in both healthy and ARDS subjects [18,19,20].

The aim of the current research is to evaluate the correlation between respiratory drive, as assessed by EAdi, and weaning success in a cohort of critically ill, mechanically ventilated patients with C-ARDS.

## 2. Materials and Methods

The study was approved by the local Ethics committee (Comitato Etico Interaziendale Milano Area 1, registration 2020/ST/207, approved date: 9th September, 2020) and written informed consent was obtained according to Italian regulations. Consecutive patients, admitted to the Intensive Care Unit (ICU) of the San Carlo Borromeo Hospital in Milan, Italy, between 8 March 2020 and 1 March 2022, were retrospectively identified.

Inclusion criteria were as follows: age > 18 years, diagnosis of C-ARDS according to clinical and radiological criteria in the Berlin Definition [21] in the presence of a SARS-CoV-2 infection, as confirmed by an RT-PCR on nasopharyngeal swab [22], absence of intravenous sedation with a level of sedation on the Richmond Agitation Sedation Scale of 0 or -1, difficult weaning from mechanical ventilation, and the presence of EAdi monitoring via EAdi nasogastric tube (by Servo-i or Servo-u mechanical ventilators equipped for EAdi monitoring, Maquet Critical Care, Solna, Sweden) with at least 6 recordings of EAdi values on consecutive days, regardless of ventilation mode. Difficult weaning was defined as patients who failed initial weaning and required at least 3 SBTs or at least 7 days from the first SBT to successful weaning [9].

All patients were deeply sedated and mechanically ventilated at ICU admission. The clinical management of patients was standardized according to local and regional suggestion [23,24]. The principle of ventilation strategy was early systematic application of the lung protective ventilatory strategy: low tidal volume; medium–high levels of PEEP; prone position if the PaO_2_/FiO_2_ ratio was <150 mmHg. Weaning was considered when a patient maintained a PaO_2_/FiO_2_ >200 mmHg during assisted ventilation, with an adequate thoraco-abdominal coordination, and performed with inspiratory pressure augmentation, as suggested [25].

Personal, demographic, anthropometric, medical history, and laboratory data were collected at ICU admission (e.g., age, gender, height, predicted and actual body weight, body-mass index, C-reactive protein, SAPS II score). Additional values were collected at the time of maximum clinical severity, including blood gas and ventilatory and respiratory mechanic parameters (PaO_2_/FiO_2_ ratio; end-tidal CO_2_; the level of positive end-expiratory pressure; airway driving pressure [26]; the static compliance of the respiratory system, and alveolar dead-space [27]). These additional parameters were recorded again in association with EAdi_peak_, starting from the time of the placement of the EAdi monitoring tube, which occurred, on average, after 17 days (IQR 9–23 days) of mechanical ventilation, and was considered as time zero (T_0_), and then after 1, 2, 3, 7, and 10 days.

To assess the correlation between EAdi and weaning success, the study population was divided into weaning success and weaning failure cohorts, compared in terms of general baseline characteristics at ICU admission (i.e., demographic, anamnestic, biochemical, and clinical characteristics), blood gas and ventilatory and respiratory mechanics parameters, both at admission and at the time of maximum severity, as well as at time 0 (EAdi tube placement) and after 1, 2, 3, 7, and 10 days.

### Statistical Analysis

Categorical variables were expressed as absolute frequency (n) and percentage (%), while continuous variables were reported as median and interquartile range (IQR), as appropriate. The following statistical tests were used: Fisher’s exact test, to compare categorical variables; Student’s t test, or its nonparametric analogue; two-sample Wilcoxon rank-sum test, to compare continuous variables depending on the distribution of the analyzed variables; one-way ANOVA for repeated measures, to compare data from different variables measured at different time steps; two-way ANOVA for repeated measures, to compare the trend of different variables measured at different time steps between the two cohorts, considering success or failure in weaning (inter-subject variable) and time (intra-subject variable) as independent categorical variables. The interaction effect between success or failure in weaning and time (combined effect of time and group) was also reported in the results, and the statistical significance of intra-subject factors was corrected using the Greenhouse–Geisser method. In cases where the interaction between variables was statistically significant, a multiple-interaction comparison was conducted according to Siegel–Tukey test. In this way, for each variable, three different *p*-values are obtained:*p*_weaning_: group effect, to compare the two cohorts.*p*_day_: time effect, to compare the trend over the days.*p*_weaning#day_: combined effect, showing the interaction of the group effect with the time effect.

To evaluate the clinical–physiological characteristics and ventilatory and respiratory mechanics parameters associated with success or failure in weaning, a logistic regression analysis was performed. To deal with the longitudinal structure of the data (patients with repeated measurements over time), an analysis was performed with a random-effect logit model for repeated measures based on each patient. The model included the effect of time as a within-subject factor, with patients fitted randomly to correct for differences in the size of the treatment effect. Odds ratios (ORs), along with their 95% confidence intervals (CIs), were computed via maximum likelihood for each variable. The factors associated with success or failure in weaning were first identified using univariate logistic regression, and were therefore included in a multiple logistic regression model. A stepwise manual procedure was used to define the final model, in which variables with *p* < 0.05 were retained. Calibration of the model was assessed by the Hosmer–Lemeshow test and discrimination ability by analysis of the area under curve (AUC) of the receiver-operating characteristics (ROC) curve.

For all the comparisons, adjusted, two-tailed P-values of less than 0.05 (*p* < 0.05) were considered statistically significant. The analysis was performed using STATA 14.0 statistical software (StataCorp LLC, College Station, TX, USA).

## 3. Results

Twenty patients were included in the current investigation. Figure 1 shows the flow chart for patient selection.

Table 1 summarizes the demographic and clinical characteristics of included subjects as well as the comparison between patients with weaning success (13, 65%) and weaning failure (7, 35%).

The only baseline variables that were shown to be statistically different between the two cohorts were age (with patients in the weaning failure group being, on average, older than those in the weaning success group) and SAPS II score (with subjects in the weaning failure group having a higher severity at ICU admission).

Table 2 reports the course of gas exchange, ventilatory and respiratory mechanics parameters in the different phases of the study in patients with weaning success or weaning failure. In summary, FiO_2_ is higher and shows an increasing trend over time in the weaning failure group, whereas it is lower and has a descending course in the weaning success group; respiratory rate shows a tendency to increase over time in the weaning failure group. As for gas exchange, subjects in the weaning failure group have significantly lower oxygenation (*p*_weaning_ < 0.0001) and higher carbon dioxide (*p*_weaning_ = 0.0001), despite similar minute ventilation (*p*_weaning_ = 0.6522), which is explained by the higher alveolar dead space (*p*_weaning_ = 0.0001).

Figure 2 shows the time course of the electrical activity of the diaphragm in the two groups. Patients in the weaning failure group had, on average, higher values of EAdi (*p*_weaning_ < 0.0001), with an upward trend over time (*p*_day_ = 0.0075), as compared with patients in the weaning success group, who had lower values which remained stable over time.

The figure shows the trend over time of the electrical activity of the diaphragm (EAdi) in patients with weaning success (white circles) or weaning failure (black circles). The comparison was performed with two-way ANOVA (see the Methods section for details). *p*_weaning_ < 0.0001; *p*_day_ = 0.0075; *p*_weaning#day_ = 0.0800.

In the multivariable logistic regression model, variables independently associated with weaning success were as follows: age (OR = 0.02 [95%CI: 0.01–0.11]), where for each additional year of age, the odds of weaning success are reduced by 98%; PaO_2_/FiO_2_ (OR = 1.10 [95%CI: 1.01–1.21]), i.e., for each additional point of PaO_2_/FiO_2_ ratio, the odds of weaning success increase by 10%; and EAdi (OR = 0.16 [95%CI: 0.08–0.34]), that is, for each additional microvolt of EAdi, the odds of weaning success are reduced by 84%. The final model had an area under the ROC curve of 0.943 [IQR 0.902–0.985] and the Hosmer–Lemeshow test resulted in *p* = 0.9367. Table 3 reports the odds ratios, determined via univariable and multivariable analyses, for the variable included in the final model.

## 4. Discussion

The main findings of this retrospective, observational investigation can be summarized as follows: in a small cohort of mechanically ventilated, critically ill patients admitted to the ICU for moderate–severe C-ARDS, with a difficult weaning from mechanical ventilation, a successful weaning was associated with a lower age, higher oxygenation and a lower respiratory drive, as assessed at the bedside by monitoring the electrical activity of the diaphragm.

### 4.1. Comparison of the Findings with Other Studies

The study population showed baseline characteristics on admission that were very similar to those of patients with ARDS from COVID-19 admitted to other Italian [28] and worldwide [29] ICU centers in terms of sex (85% male patients, in line with the available literature on C-ARDS [30]), age (median age of 66 years, only slightly higher than reported in the literature [31]), and BMI (on average, 29.8 kg/m^2^, consistent with many meta-analyses reporting average values between 28 and 30 [32]).

Various reports are available on the negative outcome and complications associated with a difficult weaning from mechanical ventilation [33,34], and support the importance of weaning failure in the management of C-ARDS: weaning failure seems to be a key factor related to the subject’s final outcome.

The role of age may appear intuitive and aligned with the recent literature on COVID-19 patients [35], as well as the available evidence regarding classic ARDS. Although burdened by limitations such as intrapulmonary shunt and dead space [36] and the nonlinear correlation with FiO_2_ [37], the PaO_2_/FiO_2_ ratio represents the best clinical index available at present to assess the entity of lung parenchymal impairment. In a multicenter investigation of C-ARDS patients, a lower PaO_2_/FiO_2_ ratio was associated with a reduced likelihood of liberation from mechanical ventilation [33], as previously demonstrated in classic ARDS [38].

Notably, we found that EAdi might represent a valid prognostic index for weaning success, as it showed significantly lower values from the beginning of monitoring and a more stable trend in the weaning success group than in the failure group, with the latter showing a progressive increase in EAdi values from the third day of monitoring onward. Respiratory drive, defined as the activity of respiratory centers [39], affects the pathophysiology and clinical outcome of ARDS [20]: indeed, ARDS patients display a discrepancy between an increase in respiratory drive and the expected increase in ventilation (which has been termed neuroventilatory dissociation), which may be related to neuromuscular impairment and altered respiratory mechanics (atelectasis and decreased lung and chest wall compliance) [40]. Of note, weaning failure subjects had both increased EAdi values and alveolar dead space, despite a similar tidal volume and respiratory rate; this might suggest the occurrence of some degree of neuroventilatory dissociation, as the increased drive could not lead to increased alveolar ventilation.

No reference range for absolute EAdi values could be established due to the high interindividual variability in signal amplitude [41]; nevertheless, the Eadi trend allows for the tracking of neural output changes in individual patients [42,43]. EAdi alterations may underlie both central (e.g., sedation, brain injury) and peripheral (e.g., phrenic nerve injury or neuromuscular junction disease) causes of respiratory drive alteration. For instance, increased EAdi may suggest an increased respiratory drive due to muscular weakness [44], while reduced EAdi may suggest a reduced respiratory drive related to over-assistance in mechanical ventilation or excessive sedation [45]. In the present study, all patients were alert, calm and cooperative, or drowsy (i.e., with values on the RASS scale ranging from 0 to −1), potentially excluding the impact of sedation on the neural respiratory drive.

### 4.2. Respiratory Drive and ARDS

ARDS (and, in all likelihood, C-ARDS) is generally associated with alterations in respiratory drive induced by mechanisms specifically associated with lung inflammation and altered respiratory mechanics. Decreased lung and chest wall compliances increase the elastic load and may alter the neuro-mechanical coupling between strain and diaphragmatic excursion. The result is a decrease in effective ventilation and an increase in PaCO_2_, which induces a stimulation of respiratory drive and a dissociation between drive and effective ventilation. Pain, anxiety, fear, and discomfort are common in patients with ARDS, and all can affect the respiratory drive [46]. Pain affects respiratory drive both through behavioral responses and through a direct reflex on respiratory centers at the level of the medulla. On the other hand, the use of sedatives may reduce respiratory drive [45]. A poor patient–ventilator interaction is another determinant of drive in mechanically ventilated ARDS subjects: asynchronies may increase respiratory drive, since they cause discomfort and increased work when breathing [47]. Moreover, the more severe the lung injury, the greater the inspiratory effort will be, reflecting the increased activation of the respiratory drive [48]. An elevated respiratory drive can be considered appropriate when the activating stimulus can be corrected by an increased ventilatory response, such as during hypercapnia and hypoxemia, when an increased ventilation is the physiological response designed to correct these alterations. In contrast, several stimuli that increase the activity of respiratory centers in ARDS are not modified by ventilatory feedback, such as inflammation, pain, and anxiety, which all inappropriately induce an elevated respiratory drive. In the case of appropriately elevated respiratory drive, treatment should facilitate the ventilatory response (e.g., by increasing ventilatory support); in contrast, an inappropriately elevated drive requires specific treatment (e.g., anxiolytic drugs). In the context of ARDS, the effects on the lung should always be monitored, and an elevated respiratory drive, whether appropriate or inappropriate, should be controlled if it results in the generation of excessive pulmonary stress, resulting in an increased risk of self-inflicted lung injury [49]. Thus, a high respiratory drive may not only correlate with the severity of ARDS but, if not carefully managed, may contribute to lung and diaphragm damage.

### 4.3. Monitoring of Respiratory Drive during the Weaning Phase

The monitoring of respiratory drive is considered essential to titrate the external ventilator support during respiratory weaning in assisted ventilation, in order to avoid both under-assistance, which would result in respiratory fatigue and exhaustion, and over-assistance, which would instead result in diaphragm dysfunction from disuse. As observed by Jonkman and coll [50], during the weaning of patients with ARDS, excessive respiratory drive and high ventilatory demands increase dyspnea and may lead to weaning failure [51]. In patients with reduced respiratory muscle strength and excessive respiratory drive, the ability of the muscle to respond to neural demands is insufficient; therefore, dyspnea is experienced as a form of excessive inspiratory effort. The activation of accessory respiratory muscles appears to be strongly correlated with the intensity of dyspnea [52] and may lead to weaning failure [53]. The EAdi signal can be used as an indicator to determine the level of ventilatory support that is needed and to optimize weaning. Generally, as soon as the patient’s condition improves, the amplitude of the EAdi signal decreases, leading to a reduction in the level of assistance provided by the ventilator. This decrease in EAdi over time has been proposed as an index to consider weaning from ventilation and extubation.

The of role diaphragmatic electromyography as a predictor of respiratory weaning failure in a population of mechanically ventilated patients was evaluated by Dres and coll. [54], who found that EAdi allows for discrimination, from the earliest stages of SBT, of weaning success or failure from mechanical ventilation, and thus constitutes a potential reliable and early predictive marker for discriminating these two categories of patients. Another study [55] found that, in a population of mechanically ventilated, critically ill patients, extubation failed more frequently in patients characterized by a higher respiratory drive, as measured by EAdi. These findings were in agreement with previous studies showing that the occlusion pressure at 100 ms (P0.1), commonly used as a proxy for neural drive, was higher in patients experiencing weaning failure [56,57]. A recent physiological study used surface electromyography in patients undergoing a spontaneous breathing trial (SBT), and found that the electrical activity of respiratory muscles increased during SBT, regardless of SBT outcome, and patients who failed the SBT had a higher increase in the activity all the inspiratory muscles compared with those who passed the SBT [58].

## 5. Limitations

The present study has several limitations, which must be taken into consideration when interpreting its results. First, some limitations are inherent in the retrospective design of the study itself, which was based on the data available in patient charts, recorded by medical and nursing staff in an, at times, emergent work setting; this caused some of the data to be missing and, in addition, might have made the data susceptible to inaccuracy. The validity of the study is further limited by the fact that it was conducted in a single center, an element that could undermine the generalizability of the results, but that, on the other hand, allowed for a similar management of the ventilator aspects, which were standardized according to the local protocols. Our case-mix, as in other similar series of COVID-19-related ARDS [28], was mainly composed of male subjects. This poses the issue of a significant gender imbalance, which might affect the external validity of our findings. Indeed, gender and sex differences are becoming increasingly recognized as important, albeit underreported and underconsidered, factors in intensive care medicine [59]. Moreover, our study lacks a control group, which could have strengthened the findings by providing a baseline for comparison. Similarly, we enrolled adult patients, and our findings cannot be directly extrapolated to the pediatric population. The last limitation is related to the relatively small sample size and the fact that, due to the inherent characteristics of the study, the patient cohort was very selective and included only patients with respiratory failure and difficult weaning. On the other hand, this study has some important strengths; chief among them is it being, to our knowledge, the first study to specifically investigate the use of monitoring EAdi as a prognostic factor for the respiratory weaning of C-ARDS patients.

## 6. Conclusions

In conclusion, in patients with C-ARDS with difficult weaning, a high respiratory drive was associated with failure of respiratory weaning, independently of age and oxygenation. In these patients, the bedside monitoring of EAdi during the weaning phase might be a useful tool to identify patients at risk of weaning failure, regardless of the mechanical ventilation mode. This would make it possible to more accurately select the optimal timing for the start of weaning from ventilation, so that only those patients with the highest probability of weaning success would be directed to it.

## Figures and Tables

**Figure 1 jcm-13-01120-f001:**
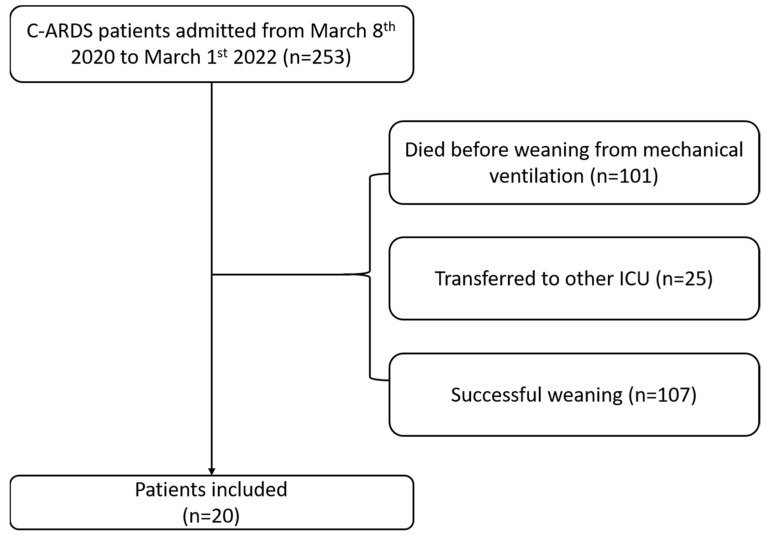
Flow chart of patient selection.

**Figure 2 jcm-13-01120-f002:**
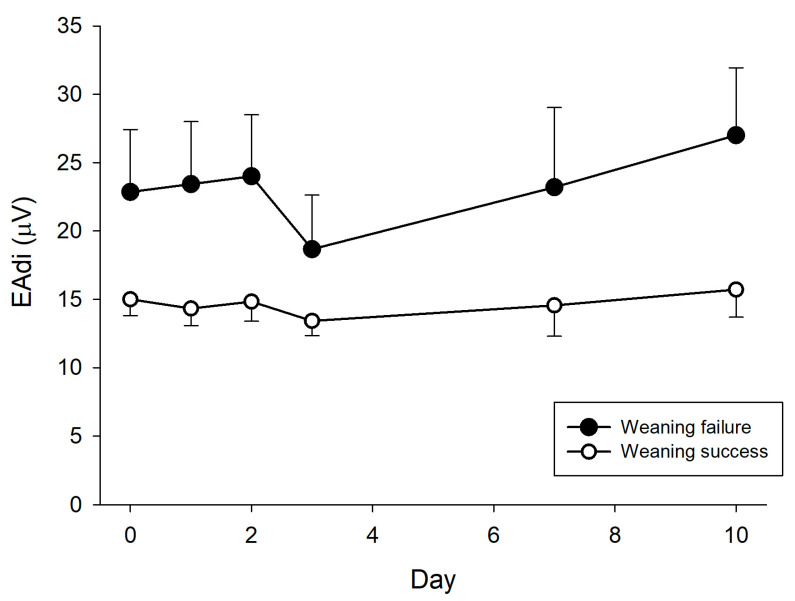
Time course of the electrical activity of the diaphragm in patients with weaning success or weaning failure.

**Table 1 jcm-13-01120-t001:** Baseline characteristics of the whole cohort upon admission to the Intensive Care Unit and comparison between patients with weaning success or weaning failure.

	Whole Cohort n = 20	Weaning Success n = 13 (65%)	Weaning Failure n = 7 (35%)	*p*-Value
**Anthropometric measures**				
Male sex (n - %)	17 (85)	11 (84.6)	6 (85.7)	0.7300
Actual body weight (kg)	83 (75–93)	82 (77.50–91)	85 (71–95)	>0.9999
Height (cm)	170 (165–175)	171 (162–175)	170 (168–172)	0.4011
Predicted body weight (kg)	54 (60–67)	64 (58–67)	64 (62–65)	0.4691
Body mass index (kg/m^2^)	29.8 (25.3–32.8)	29.7 (25.8–32.9)	30.0 (24.8–32.1)	0.9684
**Age (years)**	66 (60–69)	63 (59–68)	69 (64–74)	0.0291
**Comorbidities**				
Hypertension (n - %)	7 (35)	11 (68.80)	18 (75)	0.6650
Diabetes (n - %)	3 (15)	1 (6.25)	4 (16.70)	0.9524
COPD (n - %)	3 (15)	1 (7.7)	2 (28.6)	0.2120
AACCI (points)	3 (2–5)	3 (2–5)	4 (3–5)	0.3539
**Severity scores**				
SOFA (points)	3 (2–4)	3 (2–4)	4 (2–8)	0.2369
**SAPS II** (points)	28 (23–34)	27 (18–29)	35 (28–49)	0.0190
**Biochemical parameters**				
PCT (μg/L)	0.28 (0.19–0.76)	0.34 (0.21–0.82)	0.27 (0.18–0.34)	0.5782
CRP (mg/dL)	9.0 (2.8–14.6)	9.9 (2–17.9)	7.2 (3.7–11.8)	0.4054
Fibrinogen (mg/dL)	640 (562–674)	660 (579–674)	580 (548–690)	0.8429
D-dimer (ng/mL)	1024 (560–1666)	591 (520–1695)	1503 (1151–1638)	0.1223
Hb (g/dL)	12.1 (10.9–13.4)	12 (11–12.8)	12.2 (10.3–13.6)	0.9684
PLTs (10^3^/μL)	242 (183–299)	257 (214–331)	187 (183–257)	0.1776
WBC (10^3^/μL)	9.5 (7.4–13.6)	9.4 (6.7–13)	10.5 (8.1–16.7)	0.3620
Creatinine (mg/dL)	0.68 (0.53–0.85)	0.66 (0.62–0.81)	0.70 (0.55–0.96)	0.6917
Bilirubin (mg/dL)	0.50 (0.40–0.76)	0.40 (0.40–0.60)	0.60 (0.40–0.80)	0.1939
AST–GOT (U/L)	35 (24.50–43)	37 (29–46)	29 (19–36)	0.1776
ALT–GPT (U/L)	33.50 (22–45)	38 (25–73)	27 (16–41)	0.1220
**Ventilatory parameters**				
PEEP (cmH_2_O)	10 (10–12)	10 (10–10)	12 (10–12)	0.1511
FiO_2_ (%)	75 (57–82)	70 (60–80)	80 (50–100)	0.6591
Tv (mL)	480 (450–480)	480 (450–480)	480 (420–490)	0.7988
Tv/PBW (mL/kg)	7.27 (6.98–7.98)	7.49 (6.80–8.43)	7.27 (6.98–7.66)	0.1403
P_plat_ (cmH_2_O)	20 (18–21)	19 (18–21)	20 (16–23)	0.7682
DP (cmH_2_O)	10 (8–11)	10 (8–11)	9 (8–12)	0.6195
Crs (mL/cmH_2_O)	53 (43–57)	50 (44–56)	53 (38–61)	0.6252
pH	7.44 (7.36–7.46)	7.44 (7.37–7.46)	7.42 (7.36–7.51)	0.9682
PaCO_2_ (mmHg)	43 (37–51)	43 (37–51)	43 (41–49)	0.8423
PaO_2_ (mmHg)	99 (82–134)	99 (74–124)	131 (84–141)	0.2847
PaO_2_/FiO_2_ (mmHg)	148 (126–177)	140 (122–165)	171 (131–235)	0.1221
BE (mmol/L)	4.4 (2.1–8.9)	4.1 (2.2–5.9)	5.6 (2.1–9.1)	0.7213
EtCO_2_ (mmHg)	31 (27–37)	33 (29–38)	29 (27–34)	0.2754
**Outcomes**				
Duration of ventilation (days)	35 (13–44)	36 (8–44)	32 (30–39)	0.9053
ICU length of stay (days)	36 (13–45)	37 (9–45)	35 (30–39)	0.8740

COPD: chronic obstructive pulmonary disease; AACCI: age-adjusted Charleson comorbidity index; SOFA: sequential organ failure assessment score; SAPS II: simplified acute physiology score, 2nd version; PCT: procalcitonin; CRP: C-reactive protein; Hb: hemoglobin, PLTs: platelet count; WBC: white blood cell count; PEEP: positive end-expiratory pressure; FiO_2_: fraction of inspired oxygen; TV: tidal volume; PBW: predicted body weight; Pplat: airway plateau pressure; DP: airway driving pressure; Crs: respiratory system compliance; PaO_2_: partial pressure of oxygen in arterial blood; PaCO_2_: partial pressure of carbon dioxide in arterial blood; BE: base excess; EtCO_2_: end tidal carbon dioxide.

**Table 2 jcm-13-01120-t002:** Comparison of gas exchange, ventilatory, and respiratory mechanics parameters in the different phases of the study in patients with weaning success or weaning failure.

	Weaning Success n = 13 (65%)	Weaning Failure n = 7 (35%)	*p* _weaning_	*p* _day_	*p* _weaning#day_
**PEEP (cmH_2_O)**			0.8218	0.0880	0.7518
T_0_	10 (7–10)	10 (8–10)			
T_1_	10 (6–12)	8 (8–10)			
T_2_	10 (8–12)	10 (8–10)			
T_3_	10 (8–10)	10 (8–10)			
T_7_	8 (8–10)	8 (8–10)			
T_10_	8 (6–9)	8 (7–8)			
**FiO_2_ (%)**			<0.0001	0.7325	0.0005
T_0_	40 (40–60)	50 (45–55)			
T_1_	50 (40–60)	50 (50–60)			
T_2_	50 (40–55)	60 (50–70)			
T_3_	40 (40–60)	60 (55–70)			
T_7_	37 (35–45)	70 (60–80)			
T_10_	40 (32–40)	65 (45–80)			
**RR (1/min)**			0.3448	0.0564	0.0054
T_0_	20 (19–24)	20 (14–22)			
T_1_	23 (20–25)	24 (20–26)			
T_2_	24 (20–25)	24 (14–26)			
T_3_	22 (18–25)	26 (22–28)			
T_7_	20 (18–28)	29 (28–32)			
T_10_	25 (19–32)	24 (20–27)			
**Tv (mL)**			0.1172	0.7661	0.2139
T_0_	616 (486–715)	600 (585–631)			
T_1_	581 (537–660)	570 (457–696)			
T_2_	527 (432–636)	640 (457–660)			
T_3_	572 (511–646)	610 (457–745)			
T_7_	566 (493–655)	475 (390–550)			
T_10_	584 (468–600)	492 (380–642)			
**Tv/PBW (mL/kg)**			0.1187	0.7671	0.2746
T_0_	9.9 (8.2–10.8)	9.3 (8.7–9.8)			
T_1_	8.6 (8.2–11.1)	8.4 (7.1–10.8)			
T_2_	9.0 (6.8–10.9)	9.7 (7.1–11.4)			
T_3_	8.7 (8.0–10.4)	9.2 (7.1–12.4)			
T_7_	9.6 (8.7–9.9)	7.1 (6.0–8.0)			
T_10_	9.3 (7.7–12.5)	7.2 (5.6–9.5)			
**MVe (L/min)**			0.6522	0.8115	0.4490
T_0_	14 (13–15)	12 (9–16)			
T_1_	12 (10–15)	13 (12–14)			
T_2_	13 (10–15)	12 (9–16)			
T_3_	13 (10–14)	16 (13–18)			
T_7_	13 (12–16)	13 (12–15)			
T_10_	14 (9–18)	12 (10–11)			
**EAdi (μV)**			<0.0001	0.0075	0.0800
T_0_	15 (14–16)	20 (15–30)			
T_1_	13 (12–16)	25 (13–29)			
T_2_	14 (12–16)	24 (16–30)			
T_3_	13 (12–14)	18 (10–22)			
T_7_	13 (10–14)	25 (14–30)			
T_10_	14 (12–20)	28 (18–35)			
**PaO_2_ (mmHg)**			0.0576	0.5596	0.0465
T_0_	96 (85–104)	97 (87–114)			
T_1_	100 (91–112)	100 (83–119)			
T_2_	102 (96–107)	109 (70–139)			
T_3_	90 (80–119)	90 (65–117)			
T_7_	99 (77–127)	78 (65–91)			
T_10_	105 (98–118)	74 (69–91)			
**PaCO_2_ (mmHg)**			<0.0001	0.5577	0.6513
T_0_	41 (36–45)	43 (38–57)			
T_1_	39 (38–45)	47 (38–50)			
T_2_	40 (36–45)	48 (40–62)			
T_3_	40 (37–45)	47 (39–52)			
T_7_	44 (36–50)	54 (43–56)			
T_10_	45 (36–48)	53 (53–55)			
**PaO_2_/FiO_2_ (mmHg)**			<0.0001	0.7632	0.0001
T_0_	210 (125–260)	185 (167–228)			
T_1_	224 (157–280)	189 (166–200)			
T_2_	258 (185–291)	161 (151–232)			
T_3_	218 (150–291)	148 (105–213)			
T_7_	262 (220–302)	111 (89–163)			
T_10_	298 (224–353)	136 (92–191)			
**Crs (mL/cmH_2_O)**			0.2431	0.0987	0.3311
T_0_	32 (28–40)	39 (26–43)			
T_1_	35 (28–40)	38 (27–46)			
T_2_	31 (30–35)	41 (26–48)			
T_3_	36 (26–40)	33 (25–40)			
T_7_	34 (28–36)	26 (23–38)			
T_10_	31 (27–39)	27 (21–34)			
**EtCO_2_ (mmHg)**			0.1738	0.2703	0.3215
T_0_	30 (28–35)	26 (24–40)			
T_1_	27 (26–37)	29 (26–35)			
T_2_	30 (26–35)	32 (28–34)			
T_3_	30 (25–35)	31 (26–36)			
T_7_	35 (30–37)	33 (29–35)			
T_10_	33 (31–38)	38 (37–40)			
**Vd/Vt (%)**			0.0001	0.4257	0.7019
T_0_	25.0 (19.3–29.2)	37.5 (29.8–43.6)			
T_1_	25.0 (22.5–32.5)	32.4 (25.5–36.5)			
T_2_	25.7 (16.6–33.3)	32.0 (26.1–33.3)			
T_3_	26.9 (17.7–32.4)	34.1 (33.3–41.6)			
T_7_	20.4 (17.3–28.2)	38.8 (30.9–41.0)			
T_10_	23.4 (14.5–24.4)	28.0 (25.2–32.0)			

PEEP: positive end-expiratory pressure; FiO_2_: fraction of inspired oxygen; RR: respiratory rate; TV: tidal volume; PBW: predicted body weight; MVe: minute ventilation; EAdi: electrical activity of the diaphragm; PaO_2_: partial pressure of oxygen in arterial blood; PaCO_2_: partial pressure of carbon dioxide in arterial blood; Crs: respiratory system compliance; EtCO_2_: end tidal carbon dioxide; Vd/Vt: Alveolar dead space.

**Table 3 jcm-13-01120-t003:** Odds ratios upon univariable and multivariable analyses for the variable included in the final model.

Variable	Univariate	Multivariate
Age	OR 0.60 (95%CI 0.52–0.78)	OR 0.02 (95%CI 0.01–0.11)
PaO_2_/FiO_2_	OR 1.02 (95%CI 1.01–1.11)	OR 1.10 (95%CI 1.01–1.21)
EAdi	OR 0.86 (95%CI 0.81–0.93)	OR 0.16 (95%CI 0.08–0.34)

FiO_2_: fraction of inspired oxygen; PaO_2_: partial pressure of oxygen in arterial blood; EAdi: electrical activity of the diaphragm.

## Data Availability

The data presented in this study are available on request from the corresponding author.

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
