# Peer review of "A High Respiratory Drive Is Associated with Weaning Failure in Patients with COVID-19-Associated Acute Respiratory Distress Syndrome: The Role of the Electrical Activity of the Diaphragm"

_jcm, 2024, doi:10.3390/jcm13041120_

Round 1

Reviewer 1 Report

Comments and Suggestions for Authors

The article titled "A high respiratory drive is associated with weaning failure in patients with COVID-19-associated ARDS: the role of the electrical activity of the diaphragm." The study investigates the relationship between respiratory drive, as indicated by the electrical activity of the diaphragm (EAdi), and the success or failure of weaning patients from mechanical ventilation in cases of COVID-19-associated ARDS (C-ARDS). It includes a detailed methodology, statistical analysis, and discussion of the results, comparing various parameters between patients who succeeded and failed in weaning. The study concludes that higher respiratory drive is associated with weaning failure in C-ARDS patients. The authors suggest that monitoring EAdi could be valuable in identifying patients at risk of weaning failure.  Here are some corrections

Rewrite the first sentence of introduction in easy words 

The sentence 293-296 is supported by reference [51] however, there is nothing in referenced article related to the sentence

There is no need to cite the reference 27 

Concerns 

The factors that influence the outcome of a weaned child are not taken into account in this study. The study does not mention a control group for comparison which could strengthen the findings by providing a baseline for comparison. In order to assess the robustness of these findings, it would be useful to provide more detail on the statistical analysis carried out for the purpose of deriving the odds ratio. There is a significant gender imbalance, which may affect the generalizability of the results, with 85% of the subjects being male. 

Comments on the Quality of English Language

Minor correction

Author Response

Reviewer 1

The article titled "A high respiratory drive is associated with weaning failure in patients with COVID-19-associated ARDS: the role of the electrical activity of the diaphragm."

The study investigates the relationship between respiratory drive, as indicated by the electrical activity of the diaphragm (EAdi), and the success or failure of weaning patients from mechanical ventilation in cases of COVID-19-associated ARDS (C-ARDS). It includes a detailed methodology, statistical analysis, and discussion of the results, comparing various parameters between patients who succeeded and failed in weaning. The study concludes that higher respiratory drive is associated with weaning failure in C-ARDS patients. The authors suggest that monitoring EAdi could be valuable in identifying patients at risk of weaning failure.

Here are some corrections

Rewrite the first sentence of introduction in easy words

R: modified as requested. The sentence now reads: “While most people with COVID-19 experience mild symptoms, around 15% may develop a severe condition known as COVID-19-associated acute respiratory distress syndrome (C-ARDS). This is the primary reason for ICU admission and is linked to a poor outcome.”

The sentence 293-296 is supported by reference [51] however, there is nothing in referenced article related to the sentence

R: the reference has been modified. The next now refers to: Sklienka P, Frelich M, Burša F. Patient Self-Inflicted Lung Injury-A Narrative Review of Pathophysiology, Early Recognition, and Management Options. J Pers Med. 2023 Mar 28;13(4):593. doi: 10.3390/jpm13040593.

There is no need to cite the reference 27

R: the reference was removed, as requested

Concerns

The factors that influence the outcome of a weaned child are not taken into account in this study.

R: We thank the reviewer for the suggestion that we consider factors influencing the outcome of weaned children. While it is an interesting suggestion, unfortunately, it is beyond the scope of the present work because we only enrolled adult patients. We have added a sentence that mentions this further limitation in the limitation section.

The study does not mention a control group for comparison which could strengthen the findings by providing a baseline for comparison.

R: we thank the reviewer for this interesting comment, which is a major limitation to our manuscript. As such, this has been added in the limitation section

In order to assess the robustness of these findings, it would be useful to provide more detail on the statistical analysis carried out for the purpose of deriving the odds ratio.

R: The reviewer is correct and the methods were incomplete. We added more details to the paragraph, which now reads: “To deal with the longitudinal structure of the data (patients with repeated measurements over time), the analysis was performed with a random-effect logit model for repeated measures based on each patient. The model included the effect of time as a within-subject factor, with patients fitted as random to correct for differences in the size of the treatment effect. Odds ratios (ORs), along with their 95% confidence intervals (CIs), were computed via maximum likelihood for each variable.

There is a significant gender imbalance, which may affect the generalizability of the results, with 85% of the subjects being male.

R: we would like to thank the reviewer for this insightful comment. In fact, gender and sex differences are becoming increasingly recognised as important, albeit underreported and underconsidered, factor in intensive care medicine, as recently reviewed (Merdji, H., Long, M.T., Ostermann, M. et al. Sex and gender differences in intensive care medicine. Intensive Care Med 2023; 49: 1155–1167). This consideration and the reference are now reported in the revised version of the manuscript

Reviewer 2 Report

Comments and Suggestions for Authors

The authors of the study raised a very important issue. Mechanical ventilation is the main supportive treatment for severe cases of COVID-19-related ARDS (C-ARDS). Weaning failure is common and associated with poorer outcomes. We examined the role of respiratory drive by assessing monitoring of diaphragm electrical activity (EAdi) as a predictor of weaning failure. The study included consecutive mechanically ventilated patients admitted to the ICU for C-ARDS with difficulty in weaning. The obtained conclusions allowed the author to put forward a very important therapeutic implication. In critically ill patients with moderately severe C-ARDS and difficult weaning from mechanical ventilation, successful weaning from ventilation was associated with lower age, higher oxygenation and worse respiratory function, as assessed at the bedside using EAdi monitoring. 

Author Response

The authors of the study raised a very important issue. Mechanical ventilation is the main supportive treatment for severe cases of COVID-19-related ARDS (C-ARDS). Weaning failure is common and associated with poorer outcomes. We examined the role of respiratory drive by assessing monitoring of diaphragm electrical activity (EAdi) as a predictor of weaning failure. The study included consecutive mechanically ventilated patients admitted to the ICU for C-ARDS with difficulty in weaning. The obtained conclusions allowed the author to put forward a very important therapeutic implication. In critically ill patients with moderately severe C-ARDS and difficult weaning from mechanical ventilation, successful weaning from ventilation was associated with lower age, higher oxygenation and worse respiratory function, as assessed at the bedside using EAdi monitoring.

R: we would like to thank this reviewer for his/her comments

Reviewer 3 Report

Comments and Suggestions for Authors

I am grateful to the editor for the opportunity to review the manuscript by Muttini et al. “A high respiratory drive is associated with weaning failure in patients with COVID-19-associated ARDS: the role of the electrical activity of the diaphragm.” Indeed, electromyographic activity of the diaphragm is a predictor of failure to stop mechanical ventilation, as shown in previous studies (Ref. 1, see below) and presented in recent reviews (Ref. 2, see below). A feature of the reviewed article is the study of one of the parameters of electromyographic activity of the diaphragm (EAdi) in dynamics during mechanical ventilation in patients with COVID-19-associated ARDS. The combination of these two factors determines the novelty of the study and the facts obtained in it.

The statistical analysis of the results obtained is quite sufficient and convincingly confirms the authors’ conclusions.

However, I have comments and questions for the authors of the article.

1. It is necessary to include a flowchart for patient selection in the study (how many patients with COVID-19-associated ARDS were observed in the clinic, how many of them were excluded from the study and for what reasons).

2. It is advisable to include tables with the results of univariate and multivariate logistic regression analysis in the manuscript (either in the text of the manuscript or as an appendix to the article).

3. The list of references contains many references to publications on old studies - almost half of them are on studies more than 10 years old, and even more than 20 years old.

1. Pozzi M, Rezoagli E, Bronco A, Rabboni F, Grasselli G, Foti G, Bellani G. Accessory and Expiratory Muscle Activation During Spontaneous Breathing Trial: A Physiological Study by Surface Electromyography. Front Med (Lausanne). 2022 Mar 10;9:814219. doi: 10.3389/fmed.2022.814219.

2. Bertoni M, Spadaro S, Goligher EC. Monitoring Patient Respiratory Effort During Mechanical Ventilation: Lung and Diaphragm-Protective Ventilation. Crit Care. 2020 Mar 24;24(1):106. doi: 10.1186/s13054-020-2777-y.

Comments on the Quality of English Language

No comments

Author Response

I am grateful to the editor for the opportunity to review the manuscript by Muttini et al. “A high respiratory drive is associated with weaning failure in patients with COVID-19-associated ARDS: the role of the electrical activity of the diaphragm.” Indeed, electromyographic activity of the diaphragm is a predictor of failure to stop mechanical ventilation, as shown in previous studies (Ref. 1, see below) and presented in recent reviews (Ref. 2, see below). A feature of the reviewed article is the study of one of the parameters of electromyographic activity of the diaphragm (EAdi) in dynamics during mechanical ventilation in patients with COVID-19-associated ARDS. The combination of these two factors determines the novelty of the study and the facts obtained in it. The statistical analysis of the results obtained is quite sufficient and convincingly confirms the authors’ conclusions.

However, I have comments and questions for the authors of the article.

  1. It is necessary to include a flowchart for patient selection in the study (how many patients with COVID-19-associated ARDS were observed in the clinic, how many of them were excluded from the study and for what reasons).

R: The flow chart for patient selection was included in the text as figure 1

  1. It is advisable to include tables with the results of univariate and multivariate logistic regression analysis in the manuscript (either in the text of the manuscript or as an appendix to the article).

R: A new table which reports the Odds ratios of univariate and multivariate logistic regression has been added in the text (table 3)

  1. The list of references contains many references to publications on old studies - almost half of them are on studies more than 10 years old, and even more than 20 years old.

R: we modified the references, removing some old studies and including some more recent reports, including those reported by this reviewer. We kept in the text the references to older research reports of the electrical activity of the diaphragm which we felt are needed to put our findings into context.

Round 2

Reviewer 1 Report

Comments and Suggestions for Authors

Thank you for giving chance to review the revised version, The revised version is appropriate for publication